# Prevalence and factors associated with celiac disease in high-risk patients with functional gastrointestinal disorders

**Ari Fahrial Syam** [ORCID]*[note], **Amanda Pitarini Utari**[note], **Nur Hamidah Hasanah**[‡], **Almaarif Rizky**[‡], **Murdani Abdullah**[‡]

Division of Gastroenterology, Department of Internal Medicine, Faculty of Medicine, Universitas Indonesia/Dr. Cipto Mangunkusumo Hospital, Jakarta, Indonesia

[note] These authors contributed equally to this work.
‡ NHH, AR and MA also contributed equally to this work.
* ari_syam@hotmail.com

## Abstract

Celiac disease (CD) is an autoimmune disease of the small intestine triggered by the consumption of gluten-containing foods in individuals with a genetic predisposition. CD was a rare disease until 20 years ago, when the prevalence increased. Currently, there is no data on the prevalence of CD in high-risk adult populations in Indonesia, even though there is a trend of increasing gluten consumption. Therefore, basic research is needed to determine the magnitude of CD in high-risk adult patients in Indonesia while identifying clinical signs/symptoms, illness history, and lifestyle to determine factors associated with CD in Indonesia. This study is an observational study with a cross-sectional method. Two hundred eighty-three patients who fulfilled the selection and signed the informed consent were recruited from the gastroenterology clinic of Dr. Cipto Mangunkusumo General Hospital. Patients were asked to fill out a celiac disease-related questionnaire and then given anthropometry measurement and blood test for serologic examination with ELISA, consisting of IgA anti-tissue transglutaminase (anti-TTG) and IgG anti-deaminated gliadin peptide (anti-DGP). Statistical analysis was performed using Chi-square and Multivariate logistic regression tests with SPSS software ver. 26. Statistical test differences were considered significant if the p-values were < 0.05. Eight of 283 patients are serologically confirmed with CD (2,83%). On bivariate analysis, the significant variables are age (p < 0,05), constipation (p < 0,05) and history of autoimmune disease (p < 0,05). On multivariate analysis, the only significant variable is the history of autoimmune disease (p < 0,05). This study concluded that the prevalence of CD in high-risk patients with functional gastrointestinal disorder at Dr. Cipto Mangunkusumo Hospital is relatively high (2.83%). CD-associated factors are age, constipation, and history of autoimmune disease in patients. On simultaneous interaction between these factors, autoimmune is the only significant variable associated with CD.

**Data Availability Statement:** The data may not be shared publicly as the ethical clearance (to conduct the study, including access patient's data) was

granted only for research purpose. The study data contain sensitive patient information. Queries regarding the data may be sent to the Ethics Committee of Faculty of Medicine, Universitas Indonesia at ec_fkui@yahoo.com.

**Funding:** This study is funded by the Ministry of Education, Culture, Research, and Technology (KEMENDIBUDRISTEK) of the Republic of Indonesia through the 2023 BIMA grant program, with contract number NKB- 1099/UN2.RST/HKP.05.00/2023. Authors who received the grant are AFS and APU. Funder did not play any role in the study design, data collection, and analysis, decision to publish or preparation of the manuscript.

**Competing interests:** The authors have declared that no competing interests exist.

## Introduction

Celiac disease (CD) is an autoimmune disease of the small intestine triggered by the consumption of gluten-containing foods in individuals with a genetic predisposition [1, 2]. Until the 1980s, CD was a rare disease [3]. However, in the last 20 years, the prevalence of CD has increased [2]. Globally, there was an increase in the prevalence of CD from 0.03% to 0.7% [1, 4]. In the Asia Pacific region, the prevalence of CD has yet to be discovered thoroughly [5]. There is no data on the prevalence of CD in Indonesia. However, a study reported that out of 819 chronic diarrhea patients who sought treatment at hospitals in Jakarta, five patients (0.61%) were diagnosed with CD [6].

The increasing prevalence of CD globally is related to several factors, including the development of diagnostic tests such as IgA anti-tissue transglutaminase (anti-TTG) and IgG anti-deaminated gliadin peptide (anti-DGP), whose sensitivity and specificity are high, thus more cases of CD are diagnosed [4]. Furthermore, the increase in gluten-containing food consumption such as wheat, bread, pasta, and instant noodles also occurs in Indonesia. Indonesia Grain and Feed 2018 report shows an increase in annual wheat consumption; from 22.4 kg per capita in 2015/2016 to 23 kg per capita in 2016/2017 [7]. Meanwhile, in high-risk population, the prevalence of CD was found to be more significant compared to the general population [8]. CD prevalence in high-risk populations in developing countries ranges from 1.2% to 55% [8]. The high-risk population was defined as a patient with type 1 diabetes mellitus (DM), autoimmune thyroid disease, elevated transaminases without clear etiology, symptoms of Malabsorption with chronic diarrhea, or iron deficiency anemia [2, 8–10]. Irritable bowel syndrome (IBS) patients have similar symptoms with CD; thus, further exploration of CD is needed [11]. This is coherent with an Egyptian study that showed that out of 100 IBS patients, eight (8%) met the criteria for a CD diagnosis based on anti-TTG examination and duodenal biopsy [12].

There is no data on the prevalence of CD in high-risk adult populations in Indonesia, even though there is a trend of increasing gluten consumption in Indonesia. Therefore, basic research is needed to determine the magnitude of CD in high-risk adult patients in Indonesia and identify clinical signs and symptoms, patient's disease history and lifestyle to determine factors associated with CD incidence in Indonesia.

Through this study, it is hoped that the prevalence of CD in high-risk patients with functional gastrointestinal disorders in Indonesia along with its associated factors can be identified; thus, early detection and diagnosis of CD can be carried out, and patients can be treated as early as possible. This may reduce the disease progression, complications, and economic burden at the patient's and the nation's levels. This study also provides a better understanding of the prevalence and factors associated with CD in Indonesia, thus helping to raise awareness about CD in Indonesia.

## Materials and methods

### Design and ethical approval

Our study is an observational study with a cross-sectional method. This study has been approved by the Health Research Ethic Committee (KEPK) Faculty of Medicine, Universitas Indonesia, on June 17 2019, with approval number KET-673/UN2.F1/ETIK/PPM.00.02/2019, and ethical extension number: ND-131/ UN2.F1/ETIK/PPM.00.02/2023.

### Patient recruitment

Patients were recruited from the gastroenterology clinic of Dr. Cipto Mangunkusumo General Hospital, Jakarta, from June 2023 to December 2023. Based on the sample calculation of this

study with the prevalence of celiac disease in the high-risk patient group based on the literature at 9%, generalization error set at 5%, and precision set at 2%, this study required a minimum of 280 patients. The inclusion criteria consist of (1) Patient is over or equal to 18 years old; (2) Patient is diagnosed with either irritable bowel syndrome (IBS) according to ROME IV criteria or functional diarrhea according to ROME IV criteria; (3) Patient is diagnosed with one or more conditions, as follows: type-1 diabetes mellitus, autoimmune thyroid disease, autoimmune hepatitis, dermatitis herpetiformis with gastrointestinal symptoms (nausea, vomiting, bloating, chronic diarrhea, steatorrhea), unexplained iron deficiency anemia, or total immunoglobulin-A (IgA) deficiency. The Exclusion criteria consist of (1) a Patient who is unwilling to participate in research; (2) an Uncooperative patient; (3) a Patient diagnosed with Inflammatory Bowel Disease (IBD); (4) Pregnancy; (5) a Patient has undergone abdominal surgery. Patients who fulfilled the selection criteria signed informed consent and were included in the study. Patients were asked to fill out a celiac disease-scoring questionnaire. Patients were then given blood samples for serologic examination with ELISA, consisting of IgA anti-tissue transglutaminase (anti-TTG) and IgG anti-deaminated gliadin peptide (anti-DGP). After the sample is fulfilled, hypothesis tests will be carried out on the questionnaire and serologic examination.

## Operational definitions and vables

The dependent variables consist of anti-TTG and anti-DGP examination as serologic markers for CD. Patients concluded positive CD if the anti-TTG and anti-DGP were both positively confirmed. The independent variables are collected through questionnaire, which consist of: demographic data (gender, age, and education level), anthropometry, clinical manifestation (diarrhea, constipation, palpable lump at abdomen, abdominal fullness, bowel habit changes, abdominal pain, abdominal pain reduces after defecation, abdominal pain elevates on stress, abdominal pain on menstruation, abdominal enlargement, bloating, skin lesion, headache/dizziness, arthralgia, numbness/tingling sensation, and nausea/vomiting), history of illness (CD, first-degree family with CD, autoimmune disease, first-degree family with autoimmune disease), and lifestyle (wheat consumption, frequency of wheat consumption, smoking, alcohol consumption, and physical activity).

## Statistical analysis

Statistical analysis of the categorical data was performed using the Chi-square test. The multivariate logistic regression analysis is used for variables significantly associated with CD (from the Chi-square test). Statistical test differences were considered significant if the P-values were <0.05. Analyses were performed with SPSS software v 25.5.

## Results

### Patient demographic

Eight of 283 patients are confirmed with CD, while the other 275 are not (negative CD). Most patients confirmed with celiac disease are women (6/8); this result is non-significant (p > 0,05). Most confirmed patients are aged 40–60 years old (6/8); the latter (2/6) are diagnosed in the >60-year-old age group; this result is significant (p < 0,05). Most confirmed patients are obese (4/8); the proportion of normal and obese confirmed patients is 2 (2/8) and 2 (2/8), respectively; this result is non-significant (p > 0,05). Most confirmed patients are college graduates (5/8); the proportion of elementary graduates, middle school graduates, and

**Table 1. Patient demographic and nutritional status according to CD serologic examination result.**

| Characteristic | | Confirmed patient (n = 8) | Negative Patient (n = 275) | Total sample (n = 283) | *p*-value |
|---|---|---|---|---|---|
| Gender | Man | 2 | 57 | 59 | .674 |
| | Woman | 6 | 218 | 224 | |
| Age | <40 year old | 0 | 127 | 127 | **.033\*** |
| | 40–60 year old | 6 | 105 | 111 | |
| | >60 year old | 2 | 43 | 45 | |
| BMI | Underweight | 0 | 23 | 23 | .577 |
| | Normal | 2 | 96 | 98 | |
| | Overweight | 2 | 53 | 55 | |
| | Obese | 4 | 103 | 107 | |
| Education Level | Elementary School | 1 | 7 | 8 | .483 |
| | Middle School | 1 | 12 | 13 | |
| | High School | 1 | 84 | 85 | |
| | College | 5 | 170 | 175 | |
| | Other | 0 | 2 | 2 | |

high school graduates is 1 (1/8), 1 (1/8), and 1 (1/8), respectively; this result is non-significant (p > 0,05). These results are shown in Table 1.

## Clinical manifestation

Most patients confirmed with CD deny the presence of diarrhea (6/8); the latter (2/8) confirm the presence of diarrhea; this result is non-significant (p > 0,05). All confirmed patients deny the presence of constipation (8/8); this result is significant (p < 0,05). Most confirmed patients deny the presence of a palpable lump in their abdomen (7/8); only one confirmed patient confirms this result is non-significant (p > 0,05). Most confirmed patients confirm the presence of abdominal fullness sensation (5/8); the latter (3/8) deny it; this result is non-significant (p > 0,05). Most confirmed patients confirm the presence of bowel habit change (5/8); the latter (3/8) deny it; this result is non-significant (p > 0,05). Most confirmed patients confirm the presence of abdominal pain (6/8); the latter (2/8) deny it; this result is non-significant (p > 0,05). Most confirmed patients confirm the presence of pain reduction after defecation (6/8); the latter (2/8) deny it; this result is non-significant (p > 0,05). Most confirmed patients confirm the presence of pain elevation on stress (5/8); the latter (3/8) deny it; this result is non-significant (p > 0,05). Most confirmed patients deny the presence of pain elevation on menstruation (6/8); the latter (2/8) confirm this result is non-significant (p > 0,05). Most confirmed patients deny the presence of abdominal enlargement (5/8); the last (3/8) guarantee this result is non-significant (p > 0,05). Half of the confirmed patients confirm the presence of bloating sensation (4/8); the other half (4/8) deny it; this result is non-significant (p > 0,05). Most confirmed patients deny the presence of a skin lesion (7/8); only one patient confirms; this result is non-significant (p > 0,05). Most confirmed patients deny the presence of headache or dizziness (5/8); the latter (3/8) confirm this result is non-significant (p > 0,05). Most confirmed patients deny the presence of arthralgia (5/8); the last (3/8) confirm this result is non-significant (p > 0,05). Most confirmed patients deny the presence of numbness or tingling sensation (5/8); the latter (3/8) confirm; this result is non-significant (p > 0,05). Half of confirmed patients confirm the presence of nausea and vomiting (4/8); the other half (4/8) deny it; this result is non-significant (p > 0,05). These results are shown in Table 2.

**Table 2. Patient clinical manifestation according to CD serologic examination result.**

| Clinical Manifestation | | Confirmed patient (n = 8) | Negative Patient (n = 275) | Total sample (n = 283) | *p*-value |
|---|---|---|---|---|---|
| Diarrhea | Yes | 2 | 90 | 92 | .638 |
| | No | 6 | 185 | 191 | |
| Constipation | Yes | 0 | 114 | 114 | **.023*** |
| | No | 8 | 161 | 169 | |
| A palpable lump on the abdomen | Yes | 1 | 9 | 10 | .270 |
| | No | 7 | 266 | 273 | |
| Abdominal fullness | Yes | 5 | 191 | 196 | .680 |
| | No | 3 | 84 | 87 | |
| Bowel habit changes | Yes | 5 | 159 | 164 | .790 |
| | No | 3 | 116 | 119 | |
| Abdominal pain | Yes | 6 | 223 | 229 | .676 |
| | No | 2 | 52 | 54 | |
| Pain reduces after defecation | Yes | 6 | 192 | 198 | .749 |
| | No | 2 | 83 | 85 | |
| Pain elevates stress | Yes | 5 | 147 | 152 | .611 |
| | No | 3 | 128 | 131 | |
| Pain elevates on menstruation | Yes | 2 | 62 | 64 | .872 |
| | No | 6 | 213 | 219 | |
| Abdominal enlargement | Yes | 3 | 77 | 80 | .567 |
| | No | 5 | 198 | 203 | |
| Bloating | Yes | 4 | 208 | 212 | .124 |
| | No | 4 | 67 | 71 | |
| Skin lesion | Yes | 1 | 39 | 40 | .891 |
| | No | 7 | 236 | 243 | |
| Headache/dizziness | Yes | 3 | 142 | 145 | .429 |
| | No | 5 | 133 | 138 | |
| Arthralgia | Yes | 3 | 115 | 118 | .806 |
| | No | 5 | 160 | 165 | |
| Numbness/tingling sensation | Yes | 3 | 68 | 71 | .431 |
| | No | 5 | 207 | 212 | |
| Nausea/vomit | Yes | 4 | 138 | 142 | .992 |
| | No | 4 | 137 | 141 | |

### History of illness

All patients confirmed with CD deny having a history of CD (8/8); this result is non-significant (p > 0,05). All confirmed patients also deny having a history of CD in their first-degree families (8/8); this result is non-significant (p > 0,05). Most confirmed patients confirm having a history of autoimmune disease (5/8); the latter (3/8) deny; this result is significant (p < 0,05). Most confirmed patients deny having a history of autoimmune disease in their first-degree families (6/8); the latter (2/8) confirm this result is significant (p > 0,05). These results are shown in Table 3.

### Patient lifestyle

All patients confirmed with CD have a history of wheat consumption (8/8); this result is non-significant (p > 0,05). Among these eight patients, half confirm having wheat consumption every day (4/8); two patients (2/8) confirm having wheat consumption 1–4 days/week; each

**Table 3. Patient history of Illness according to CD serologic examination result.**

| History of Illness | | Confirmed patient (n = 8) | Negative Patient (n = 275) | Total sample (n = 283) | *p*-value |
|---|---|---|---|---|---|
| Celiac disease | Yes | 0 | 1 | 1 | .811 |
| | No | 8 | 274 | 282 | |
| 1st-degree family with celiac disease | Yes | 0 | 2 | 2 | .734 |
| | No | 8 | 273 | 281 | |
| Autoimmune disease | Yes | 5 | 78 | 83 | **.050*** |
| | No | 3 | 197 | 200 | |
| 1st-degree family with autoimmune disease | Yes | 2 | 33 | 35 | .322 |
| | No | 6 | 242 | 248 | |

patient confirms having wheat consumption 1–3 days/month and <1 day/month; this result is non-significant (p > 0,05). Most confirmed patients deny having a smoking history (7/8); only one confirmed patient confirms being an active smoker; this result is non-significant (p > 0,05). All confirmed patients deny having alcohol consumption (8/8); this result is non-significant (p > 0,05). Most confirmed patients confirm having less physical activity (4/8); each of the two confirmed patients confirms having adequate physical activity (2/8) and sedentary physical activity (2/8); this result is non-significant (p > 0,05). These results are shown in Table 4.

## Multivariate analysis

Based on previous Chi-square tests, the significant independent variables were age, constipation status, and history of autoimmune disease. These three variables were then tested by binary logistic regression. From the test, it was found that the age variable had no partial effect on the incidence of CD (p > 0.05); constipation status had no partial effect on CD incidence (p > 0.05). Also, a history of autoimmune disease has no partial effect on CD (p > 0.05), as shown in Table 5.

**Table 4. Patient lifestyle according to CD serologic examination result.**

| Lifestyle | | Confirmed Patient (n = 8) | Negative Patient (n = 275) | Total sample (n = 283) | *p*-value |
|---|---|---|---|---|---|
| History of Wheat Consumption | Yes | 8 | 273 | 281 | .734 |
| | No | 0 | 2 | 2 | |
| Frequency of wheat consumption | Everyday | 4 | 86 | 90 | .605 |
| | 5–6 days/week | 0 | 18 | 18 | |
| | 1–4 days/week | 2 | 56 | 58 | |
| | 1–3 days/month | 1 | 78 | 79 | |
| | <1 day/month | 1 | 37 | 38 | |
| History of smoking | Active smoker | 1 | 18 | 19 | .400 |
| | Ex-smoker | 0 | 26 | 26 | |
| | Non-smoker | 7 | 231 | 238 | |
| History of alcohol consumption | Alcohol drinker | 0 | 3 | 3 | .604 |
| | Ex-alcohol drinker | 0 | 14 | 14 | |
| | Non-alcohol drinker | 8 | 257 | 265 | |
| Physical activity | Adequate | 2 | 63 | 65 | .750 |
| | Less | 4 | 109 | 113 | |
| | Sedentary | 2 | 103 | 105 | |

**Table 5. Multivariate analysis using binary logistic regression model.**

| Variables | Bivariate | Multivariate |
|---|---|---|
| | *p*-value | *p*-value |
| Age | .033* | .109 |
| Constipation | .023* | .996 |
| History of autoimmune disease | .050* | .035* |

## Discussion

### Patient demographic and nutritional status

Age, gender, BMI, and education were considered in this investigation. The research revealed a significant association between celiac disease (CD) and a specific age group (40–60 years old). This result aligns with a study by Ludvigsson et al., which indicated that the incidence of celiac disease is not limited to early childhood but extends across various age brackets [13]. The clinical presentation of celiac disease evolves with age, posing challenges for diagnosis in older populations [13]. The intricate connection between age and celiac disease requires further exploration to improve early detection and management, especially with the increasing recognition of late-onset cases.

While this study did not establish a significant association between gender and CD, Singh et al. reported a higher frequency of celiac disease diagnoses in women than men, with distinct clinical manifestations [14]. Gender differences in diagnosis may stem from variations in symptomatology, healthcare-seeking behaviour, and potential contributions from hormonal factors such as estrogen levels [14].

Contrary to a study by Singh et al., this investigation found no significant association between BMI and CD [14]. Singh et al. suggested that a higher BMI might increase susceptibility to celiac disease, challenging the traditional perception that the disorder predominantly affects underweight individuals [14]. Excessive adipose tissue in individuals with higher BMI leads to elevated adipokines, which modulates the immune response [14]. Additionally, obesity could impact gut microbiota composition, influencing immune system regulation [14]. The gut microbiota, comprising various microorganisms, plays a crucial role in shaping the host immune response, with a key mechanism promoting T cell and B cell development and function [14].

The study also explored the association between education level and CD, finding no significant correlation. However, specific studies directly linking education level to celiac disease are currently unavailable. Notably, a study by Roy et al. indicated no apparent relationship between CD and socioeconomic status [15].

### Clinical manifestation

Gastrointestinal (GI) related clinical manifestations consisting of nausea and/or vomiting, bowel habit changes, diarrhea, constipation, bloating, abdominal pain, abdominal enlargement, and abdominal lump were considered in this study. Among these, only constipation has a strong association with CD ($p < 0,05$), with most CD-confirmed patients (7/8) denying the presence of constipation. However, constipation may occur in CD through several factors: inflammation and damage to the intestinal lining, altered motility, nutrient malabsorption, and changes in gut microbiota [16–18]. Celiac disease causes inflammation in the small intestine, damaging the villi and causing changes in bowel habits, including constipation [16]. The inflammation and damage to the intestine can affect the normal rhythmic contractions

(peristaltic) of the muscles in the digestive tract, leading to a slowdown in the movement of stool through the intestines, contributing to constipation [17, 18]. Malabsorption of nutrients, including fats, can reduce stool volume and contribute to constipation [18]. Imbalances in the gut microbiota have been observed in individuals with celiac disease [18]. These changes may influence bowel habits, including constipation [18].

This result also contrasts with a study by Ludvigsson et al., which stated that diarrhea, steatorrhea, weight loss, or growth failure are required signs of classical CD [13]. This study also stated that atypical CD patients may appear with the following signs/symptoms: IBS symptoms, liver dysfunction, failure to thrive, thyroid dysfunction, neurologic disorder including depression and gluten ataxia, reproductive disease including abnormal menarche and/or menopause, oral/cutaneous disease including dermatitis herpetiformis, nutritional deficiency (iron deficiency) and skeletal finding [13]. Furthermore, several studies classify CD signs and symptoms into gastrointestinal, extraintestinal, silent, and asymptomatic manifestations [19, 20]. Gastrointestinal manifestations are classical and atypical: diarrhea, abdominal pain, bloating, weight loss, and constipation, dyspepsia, and IBS-like symptoms, respectively [19, 20]. Extraintestinal symptoms include dermatitis herpetiformis, osteoporosis, arthritis, and infertility [19, 20]. Neurological symptoms such as peripheral neuropathy, ataxia, and seizures have also been reported [19]. In silent and asymptomatic manifestations, individuals with celiac disease may not present with any noticeable symptoms, leading to a silent or asymptomatic form of the condition [19]. These cases are often detected through screening programs or during the evaluation of other medical conditions [19]. Uniquely, pediatric patients can present with growth failure, delayed puberty, and developmental delays [19].

## History of illness

History of CD, history of autoimmune disease, history of CD in first-degree family, and history of autoimmune disease in first-degree family were considered in this study. However, a history of autoimmune disease is the only significant factor ($p < 0,05$) associated with CD. Celiac disease is strongly associated with a genetic predisposition, and the primary genetic factor implicated in the condition's development is specific human leukocyte antigen (HLA) class II alleles [21]. The most notable alleles linked to celiac disease are HLA-DQ2 and HLA-DQ8, and individuals carrying one or both of these alleles are at an increased risk of developing celiac disease [21]. HLA-DQ2 and HLA-DQ8 are strongly associated with celiac disease, although not all individuals carrying these alleles develop the condition [21]. The exact mechanism by which these HLA alleles contribute to the autoimmune predisposition in celiac disease involves the presentation of gluten peptides to T cells within the small intestine [21]. HLA-DQ2 or HLA-DQ8 molecules expressed on the surface of APCs bind to specific gluten peptides [21]. This binding is crucial for the activation of T cells [21]. Then, T cells recognize the gluten peptides presented by the HLA-DQ molecules [21]. In genetically susceptible individuals, this recognition leads to the activation of T cells [21]. Activated T cells trigger an immune response, producing pro-inflammatory cytokines and other immune mediators [21]. This inflammatory cascade damages the lining of the small intestine [21]. The chronic inflammation and tissue damage in the small intestine result in an autoimmune response against the intestinal cells, particularly the villi [21]. Over time, the autoimmune response causes villous atrophy, a characteristic feature of CD [21]. Villous atrophy impairs nutrient absorption and leads to the clinical manifestations of celiac disease, including gastrointestinal and extraintestinal symptoms [21].

However, this result is quite different from a few studies which stated that individuals with a first-degree family history of autoimmune diseases are at an increased risk of developing

celiac disease; this is due to the inheritance of the primary genetic risk factor associated with celiac disease: the presence of HLA-DQ2 and HLA-DQ8 [22, 23]. Previous CD history of either patient or their families were non-significantly associated with CD. This may result from limited screening and diagnosis resources and a low level of knowledge and awareness about CD; thus, it is scarce for a patient who has been diagnosed with CD before.

## Patient lifestyle

As for lifestyle, history of wheat consumption, frequency of wheat consumption, history of smoking, history of alcohol consumption, and physical activity were considered in this study. No variable has a significant association with CD (p > 0,05). On the contrary, few studies have found the exact relationship between wheat consumption and CD; CD is a chronic immune-mediated disorder characterized by an abnormal response to gluten, a protein found in wheat and related grains [24]. Gluten, a complex mixture of proteins present in wheat, barley, and rye, is the primary trigger for celiac disease [24]. Gliadin, a component of gluten, is resistant to complete digestion in the small intestine in individuals with celiac disease, leading to the formation of immunogenic peptides [24]. Eliminating gluten from the diet helps alleviate symptoms, promotes intestinal healing, and prevents long-term complications [24].

Some studies have explored potential links between smoking, alcohol consumption, and celiac disease [25, 26]. They suggested a potential protective effect of smoking on the development of celiac disease, while others have not found a significant association [25, 26]. However, it is crucial to emphasize that the potential health risks associated with smoking far outweigh any speculative protective effects, and smoking cessation is always recommended for overall health [25, 26]. Alcohol itself is generally considered safe for individuals with celiac disease, as it does not contain gluten [25, 26]. However, alcoholic beverages may sometimes be made from grains that contain gluten. Individuals with celiac disease need to choose alcoholic beverages carefully and opt for those that are gluten-free [25, 26].

Research on the direct association between physical activity and celiac disease is not as extensive as studies on other aspects of celiac disease, such as genetics, symptoms, or dietary factors. However, regular physical activity has been shown to improve the overall well-being and quality of life in individuals with chronic conditions, including autoimmune disorders like celiac disease [27]. Celiac disease also affects bone health due to the Malabsorption of nutrients, especially calcium and vitamin D [28]. Exercise, particularly weight-bearing, can contribute to maintaining bone density and strength [28]. Exercise also has a modulatory effect on the immune system [29]. It also enhances the activity of various immune cells, such as natural killer cells, neutrophils, and macrophages, which play crucial roles in defending the body against pathogens [29]. Exercise also promotes better circulation, which allows immune cells to move more efficiently throughout the body [29]. Improved circulation ensures immune cells reach their target areas more effectively to combat infections [29]. Regular exercise may also influence the production and activity of immunoglobulins, such as immunoglobulin A (IgA), which plays a role in mucosal immunity, protecting the respiratory and gastrointestinal tracts [29]. Engaging in physical activity can positively impact mental health and reduce stress [30].

## Conclusion

The prevalence of CD in high-risk patients with functional gastrointestinal disorders at Dr. Cipto Mangunkusumo Hospital is relatively high (2,83%) compared to a similar study, which stated 0,61%. CD-associated factors are age, constipation, and history of autoimmune disease

in patients. On simultaneous interaction between these factors, autoimmune is the only significant variable associated with CD.

## Acknowledgments

The authors thank Aan Santi and Titis Puspitarini for their advice on statistical analysis of this study.

## Author Contributions

**Conceptualization:** Ari Fahrial Syam, Murdani Abdullah.

**Funding acquisition:** Ari Fahrial Syam, Amanda Pitarini Utari.

**Investigation:** Ari Fahrial Syam, Amanda Pitarini Utari, Murdani Abdullah.

**Methodology:** Ari Fahrial Syam, Murdani Abdullah.

**Project administration:** Ari Fahrial Syam.

**Resources:** Ari Fahrial Syam, Amanda Pitarini Utari.

**Supervision:** Ari Fahrial Syam, Amanda Pitarini Utari, Murdani Abdullah.

**Validation:** Ari Fahrial Syam, Amanda Pitarini Utari.

**Writing – original draft:** Ari Fahrial Syam, Nur Hamidah Hasanah, Almaarif Rizky.

**Writing – review & editing:** Ari Fahrial Syam, Nur Hamidah Hasanah.

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
