## [Decision Letter · Decision Letter 0]

4 Feb 2024

PONE-D-24-00819Prevalence and Factors Associated With Celiac Disease in High-Risk Patients with Functional Gastrointestinal Disorders.PLOS ONE

Dear Dr. Ari Fahrial Syam,

Thank you for submitting your manuscript to PLOS ONE. After careful consideration, we feel that it has merit but does not fully meet PLOS ONE’s publication criteria as it currently stands. Therefore, we invite you to submit a revised version of the manuscript that addresses the points raised during the review process.

Thank you for the study. I invite you to resubmit your manuscript after addressing two reviewers’ comments. When resubmitting your manuscript, please carefully consider all issues mentioned in the reviewers' comments, outline every change made point by point, and provide suitable rebuttals for any comments not addressed.

We look forward to receiving your revised manuscript.

Kind regards,

Yasin Sahin

Academic Editor

PLOS ONE

Journal Requirements:

"This study is funded by the Ministry of Education, Culture, Research, and Technology (KEMENDIBUDRISTEK) of the Republic of Indonesia through the 2023 BIMA grant program, with contract number NKB-1099/UN2.RST/HKP.05.00/2023. The authors thank Aan Santi and Titis Puspitarini for their advice on statistical analysis of this study."

"This study is funded by the Ministry of Education, Culture, Research, and

Technology (KEMENDIBUDRISTEK) of the Republic of Indonesia through the

2023 BIMA grant program, with contract number NKB-

1099/UN2.RST/HKP.05.00/2023. Authors who received the grant are AFS and APU. Funder did not play any role in the study design, data collection, and analysis, decision to publish or preparation of the manuscript."

3. In the online submission form, you indicated that [Data cannot be shared publicly and are available upon request.]. 

Additional Editor Comments:

Thank you for the study. I invite you to resubmit your manuscript after addressing two reviewers’ comments. When resubmitting your manuscript, please carefully consider all issues mentioned in the reviewers' comments, outline every change made point by point, and provide suitable rebuttals for any comments not addressed.

Reviewers' comments:

Reviewer's Responses to Questions

**Comments to the Author**

1. Is the manuscript technically sound, and do the data support the conclusions?

Reviewer #1: Partly

Reviewer #2: Yes

2. Has the statistical analysis been performed appropriately and rigorously? 

Reviewer #1: I Don't Know

Reviewer #2: Yes

3. Have the authors made all data underlying the findings in their manuscript fully available?

Reviewer #1: Yes

Reviewer #2: Yes

4. Is the manuscript presented in an intelligible fashion and written in standard English?

Reviewer #1: Yes

Reviewer #2: Yes

5. Review Comments to the Author

Reviewer #1: The authors have documented the prevalence of celiac disease (CD) in a cohort of "high-risk" patients with functional GI disorders (FGID).

Comments:

1. While the authors have attributed the reported increasing prevalence of CD to better tests availability as well as increasing gluten consumption, it must be clarified that the gross quantum of gluten consumption per person has not been implicated in the pathogenesis of CD -- it may be due to more persons consuming gluten. Besides, better tests availability and awareness are more important factors

2. FGID and IBS are heterogeneous conditions, with patients presenting with a wide spectrum of symptoms. It would have been better if the authors selected either the general population or a subgroup with IBS-d. That would have made symptom analysis more meaningful

3. FGID / IBS by definition are low-risk conditions; the presence of high-risk features should warn against such diagnoses. Do the authors mean that these patients with FGID / IBS had other features (independent of FGID / IBS) that placed them at higher possibility of a diagnosis of CD? That is reasonable -- in the final analysis, only autoimmune disease correlated with the possibility of CD

4. Do the authors have an explanation why patients with CD were more likely to be aged 40-60 y rather than younger (which is expected)?

Reviewer #2: Dear Editor,

I should first thank for inviting me as potential reviewer to read and comment on paper entitled ‘’Prevalence and Factors Associated With Celiac Disease in High-Risk Patients with Functional Gastrointestinal Disorders.

In the current study, the authors aimed to conduct a comprehensive 65 review of existing literature pertaining to this specific topic and to elucidate the role of the 66 autoantibodies in its pathogenesis.

The main title accurately reflects the major topic and content of the study.

The abstract summarizes and reflects the work described in the manuscript. Also, the abstract presents the significant points related to the background, objectives, materials and methods, results and conclusions.

The materials and methods sufficiently described for the results and conclusions that are presented in the preceding sections. The study type and design were defined in the section of the materials and methods. Inclusion and exclusion criteria are well defined. Figures and tables are sufficient and well-organized. Ethics Committee approval was received. So, the section materials and methods is adequate.

The statistical methods used are appropriate.

The section of the discussion is well organized. The conclusions are drawn appropriately supported by the literature. The manuscript adequately describes the background, present status and significance of the study. The manuscript interprets the findings adequately and appropriately, highlighting the key points clearly.

I think that it will contribute to the literature. Generally, the manuscript appropriately cites the important and authoritative references but did not cite the recent published articles. I have some major criticisms.

- According to ESPGHAN updated guidelines, IgA anti-tissue transglutaminase (anti-TTG) is recommended but IgG anti-deaminated gliadin peptide (antiDGP) is not recommended. Why did the authors use IgG anti-deaminated gliadin peptide (antiDGP) serologic test in high risk population for celiac disease? Please explain it.

- The section of the introduction and discussion is not well organized. The manuscript did not cite the recent published articles. If the recent published articles are cited and discussed in the manuscript, it would be better. There are many articles on this subject, the authors can benefit from them.

-The manuscript appropriately cites the important and authoritative references but does not cite the recent published articles. If the recent published article about celiac disease for example ‘’ Celiac disease in children: A review of the literature. World J Clin Pediatr. 2021 Jul 9;10(4):53-71’’ are cited, the manuscript would be better.

-In the current study, the authors did not cite the recent published articles. If the recent published articles about prevalence of CD in high risk population for example ‘Prevalence of celiac disease in children with type 1 DM in the South of Turkey. Iran J Pediatr 2020;30:e97306’’ and ‘’The frequency of celiac disease in children with autoimmune thyroiditis. Acta Gastroenterol Belg. 2018 Jan-Mar;81(1):5-8’’and ‘’The frequency of celiac disease in children with chronic constipation. Kuwait Medical Journal 2021; 53 (4): 383 – 387’’ are cited, the manuscript would be better.

-If the recent published articles about HLA-DQ2 and DQ8 in celiac patients for example ‘’Frequency of celiac disease and distribution of HLA-DQ2/DQ8 haplotypes among siblings of children with celiac disease. World J Clin Pediatr 2022 Jul 9;11:351-359’’ are cited, the manuscript will be better

After careful revision, I think that it will contribute to the literature.

6. PLOS authors have the option to publish the peer review history of their article (what does this mean?). If published, this will include your full peer review and any attached files.

Reviewer #1: No

Reviewer #2: No

---

## [Author Response · Author response to Decision Letter 0]

3 May 2024

Response to Editor & Reviewers

By Academic editor

1 Please ensure that your manuscript meets PLOS ONE's style requirements, including those for file naming. The PLOS ONE style templates can be found at 

Response: Thank you. We have applied your instruction on the revised manuscript.

2 funding information should not appear in the Acknowledgments section or other areas of your manuscript. We will only publish funding information present in the Funding Statement section of the online submission form. 

Please remove any funding-related text from the manuscript and let us know how you would like to update your Funding Statement. 

Response: Thank you. We have removed funding information on our revised manuscript.

3 In the online submission form, you indicated that [Data cannot be shared publicly and are available upon request.]. 

Response: After careful consideration, we have agreed that our data remains and cannot be shared publicly and made available upon request. This is due to our commitment to our ethics board, which firmly believes data obtained upon conducting research is as private and confidential as a patient’s medical record. However, once it is needed for research/audit purposes we are ready to provide it as soon as possible.

4 When completing the data availability statement of the submission form, you indicated that you will make your data available on acceptance. We strongly recommend all authors decide on a data sharing plan before acceptance, as the process can be lengthy and hold up publication timelines. Please note that, though access restrictions are acceptable now, your entire data will need to be made freely accessible if your manuscript is accepted for publication. This policy applies to all data except where public deposition would breach compliance with the protocol approved by your research ethics board. If you are unable to adhere to our open data policy, please kindly revise your statement to explain your reasoning and we will seek the editor's input on an exemption. Please be assured that, once you have provided your new statement, the assessment of your exemption will not hold up the peer review process.

Response: We are on the process of sorting and planning the data sharing. Once it is needed for research/audit purposes we are ready to provide it as soon as possible.

By Reviewer 1

1 While the authors have attributed the reported increasing prevalence of CD to better tests availability as well as increasing gluten consumption, it must be clarified that the gross quantum of gluten consumption per person has not been implicated in the pathogenesis of CD -- it may be due to more persons consuming gluten. Besides, better tests availability and awareness are more important factors

Response: Agree, thank you for your correction. We have corrected our previous statement in our revised manuscript. Related to test availability and patient awareness, we have mentioned it in discussion section, part: history of illness.

2 FGID and IBS are heterogeneous conditions, with patients presenting with a wide spectrum of symptoms. It would have been better if the authors selected either the general population or a subgroup with IBS-d. That would have made symptom analysis more meaningful

Response: Exactly, however, our study focus is celiac disease and celiac clinical manifestations are not limited to IBS but also another kind of FGID such as functional diarrhea. We would like to capture celiac patients as variation as possible.

3 FGID / IBS by definition are low-risk conditions; the presence of high-risk features should warn against such diagnoses. Do the authors mean that these patients with FGID / IBS had other features (independent of FGID / IBS) that placed them at higher possibility of a diagnosis of CD? That is reasonable -- in the final analysis, only autoimmune disease correlated with the possibility of CD 

Response: Exactly, that was our study focus (CD prevalence on high risk FGID patients), and this topic was also the first to be conducted in Indonesia.

4 Do the authors have an explanation why patients with CD were more likely to be aged 40-60 y rather than younger (which is expected)?

Response: Studies of serial serum samples have reported loss of gluten tolerance as aging start on 35-40 years old. This is may be the cause CD serologic result are most prominent at age >40 years old in our study. However, CD can develop at any age; the earlier gluten exposure start, the earlier CD develop. There are differences in clinical manifestation between children and adult CD patients. In children, most CD manifest as diarrhea, weight loss and growth impairment. While in adults, most CD manifest as constipation, steatorrhea, obesity and micronutrients deficiency. Thank you for the question, we have added this topic on discussion section of revised manuscript.

By Reviewer 2

1 According to ESPGHAN updated guidelines, IgA anti-tissue transglutaminase (anti-TTG) is recommended but IgG anti-deaminated gliadin peptide (antiDGP) is not recommended. Why did the authors use IgG anti-deaminated gliadin peptide (antiDGP) serologic test in high risk population for celiac disease? Please explain it. 

Response: We remain to use anti-DGP in our study as anti-DGP has proven to has higher specificity than anti-TTG towards CD, while anti-TTG has higher sensitivity than anti-DGP. The sensitivity, specificity, and diagnostic accuracy of the IgA anti-tTG test were 86.3%, 50.0%, and 68.6%, respectively, and those of the IgG anti-DGP test were 95.4%, 85.7%, and 90.7%, respectively. Although, in any circumstances, the gold standard examination for CD remain biopsy of duodenum with suggestive finding: villous atrophy.

References:

- Anbardar, M. H., Haghighi, F. G., Honar, N., & Zahmatkeshan, M. (2022). Diagnostic Value of Immunoglobulin G Anti-Deamidated Gliadin Peptide Antibody for Diagnosis of Pediatric Celiac Disease: A Study from Shiraz, Iran. Pediatric gastroenterology, hepatology & nutrition, 25(4), 312–320. https://doi.org/10.5223/pghn.2022.25.4.312

- Jafari, S. A., Alami, A., Sedghi, N., Kianifar, H., Kiani, M. A., Khalesi, M., & Derafshi, R. (2023). Diagnostic accuracy of anti-DGP (IgG) for celiac disease. Journal of family medicine and primary care, 12(1), 42–46. https://doi.org/10.4103/jfmpc.jfmpc_326_22

2 The section of the introduction and discussion is not well organized. The manuscript did not cite the recent published articles. If the recent published articles are cited and discussed in the manuscript, it would be better. There are many articles on this subject, the authors can benefit from them.

Response: Thank you, we have reorganized introduction and discussion section, also added several more recent publication.

3 The manuscript appropriately cites the important and authoritative references but does not cite the recent published articles. If the recent published article about celiac disease for example ‘’ Celiac disease in children: A review of the literature. World J Clin Pediatr. 2021 Jul 9;10(4):53-71’’ are cited, the manuscript would be better.

Response: Thank you for your suggestion. We have added several more recent publication.

4 In the current study, the authors did not cite the recent published articles. If the recent published articles about prevalence of CD in high risk population for example ‘Prevalence of celiac disease in children with type 1 DM in the South of Turkey. Iran J Pediatr 2020;30:e97306’’ and ‘’The frequency of celiac disease in children with autoimmune thyroiditis. Acta Gastroenterol Belg. 2018 Jan-Mar;81(1):5-8’’and ‘’The frequency of celiac disease in children with chronic constipation. Kuwait Medical Journal 2021; 53 (4): 383 – 387’’ are cited, the manuscript would be better.

Response: Thank you for your suggestion. We have added several more recent publication.

5 If the recent published articles about HLA-DQ2 and DQ8 in celiac patients for example ‘’Frequency of celiac disease and distribution of HLA-DQ2/DQ8 haplotypes among siblings of children with celiac disease. World J Clin Pediatr 2022 Jul 9;11:351-359’’ are cited, the manuscript will be better. 

Response: Thank you for your suggestion. We have added several more recent publication.

---

## [Editor Report · Decision Letter 1]

8 May 2024

Prevalence and Factors Associated With Celiac Disease in High-Risk Patients with Functional Gastrointestinal Disorders.

PONE-D-24-00819R1

Dear Dr. Ari Fahrial Syam,

We’re pleased to inform you that your manuscript has been judged scientifically suitable for publication and will be formally accepted for publication once it meets all outstanding technical requirements.

Kind regards,

Yasin Sahin

Academic Editor

PLOS ONE

Additional Editor Comments (optional):

Thank you for the study. The authors did an appropriate point-by-point response to the reviewers. I think that it will contribute to the literature.
---

## [Editor Report · Acceptance letter]

13 May 2024

PONE-D-24-00819R1 

PLOS ONE

Dear Dr. Syam, 

I'm pleased to inform you that your manuscript has been deemed suitable for publication in PLOS ONE. Congratulations! Your manuscript is now being handed over to our production team.

Kind regards, 

on behalf of

Dr. Yasin Sahin 

Academic Editor

PLOS ONE